# Impact of the change in the antitubercular regimen from three to four drugs on cure and frequency of adverse reactions in tuberculosis patients from Brazil: A retrospective cohort study

**María B. Arriaga**[1,2,3,4]*, **Ninfa M. C. Torres**[1,5], **Nelia C. N. Araujo**[1,2], **Simone C. C. Caldas**[2], **Bruno B. Andrade**[2,3,4,6,7,8]*, **Eduardo M. Netto**[1,2]

**1** Universidade Federal da Bahia, Salvador, Brazil, **2** Instituto Brasileiro de Investigação da Tuberculose, Salvador, Brazil, **3** Instituto Gonçalo Moniz, Fundação Oswaldo Cruz, Salvador, Brazil, **4** Multinational Organization Network Sponsoring Translational and Epidemiological Research (MONSTER) Initiative, Salvador, Brazil, **5** Universidad Militar Nueva Granada, Bogotá, Colombia, **6** Curso de Medicina, Faculdade de Tecnologia e Ciências (FTC), Salvador, Brazil, **7** Curso de Medicina, Universidade Salvador (UNIFACS), Salvador, Brazil, **8** Escola Bahiana de Medicina e Saúde Pública, Salvador, Brazil

* mbag711@gmail.com (MBA); bruno.andrade@fiocruz.br (BBA)

## Abstract

### Background

The Ministry of Health in Brazil included ethambutol in the intensive phase of sensible tuberculosis (TB) treatment in March 2010, due to the increasing drug resistance, and implemented the fixed dose combination in the TB treatment guidelines.

### Methods

A retrospective cohort study was performed to determine the impact of change from three to four drugs schemes on the TB cure and frequency of adverse drug reactions (ADRs) in TB patients. To answer this question, we used data from 730 randomly selected patients who received anti-TB treatment between January 2007 and December 2014 in a reference center from Salvador, Brazil.

### Findings

TB patients who received the RHEZ regimen (n = 365) developed ADRs more frequently than those treated with the RHZ (n = 365) (86 [23.6%] vs. 55 [15.1%]; p = 0.01). This difference in ADR incidence was even higher in patients above 30 years-old (64 [74.4%] vs. 36 [65.5%]; p = 0.01). The overall number of ADR episodes was greater in patients from the RHEZ group than in the group that received RHZ (170 [61.4%] vs. 107 [38.6%]; p = 0.03). Multivariable logistic regression analysis adjusted for age, alcohol use and diabetes demonstrated that patients receiving the RHEZ regimen had increased odds of developing ADRs than those undertaking the RHZ scheme (odds ratio [OR]: 1.61, 95% confidence interval

**Data Availability Statement:** All relevant data are within the paper and its Supporting Information files.

**Funding:** This study was supported by the Intramural Research Program from Fundação José Silveira, Universidade Federal da Bahia and Program of Fundação Oswaldo Cruz (FIOCRUZ). BBA is a senior researcher from the Conselho Nacional de Desenvolvimento Científico e Tecnológico (CNPq). MBA received a fellowship from the Fundação de Amparo à Pesquisa da Bahia (FAPESB). The funders had no role in study design, data collection and analysis, decision to publish, or preparation of the manuscript.

**Competing interests:** The authors have declared that no competing interests exist.

[CI]: 1.10–2.35; p = 0.015). The overall cure rate was similar between the distinct treatment groups.

## Conclusion

The patients treated with the four-drug regimen exhibited increased risk of ADRs compared to those who received the three-drug regimen, and especially in patients older than 30 years of age.

## Introduction

Approximately one quarter of the world's population is estimated to be infected with *Mycobacterium tuberculosis*, resulting in a major global health problem. In 2016, Brazil reported 2.7/100,000 deaths, and a total incidence of 41/100,000 [1]. Since 1994, the World Health Organization (WHO), together with the International Union Against Tuberculosis and Lung Disease (IUATLD), recommended the use of fixed dose combination (FDC) for TB treatment arguing that this approach simplifies drug dispensing and prevents the development of drug resistance [2]. In 2003, the WHO recommended the addition of ethambutol (E) in basic TB treatment with rifampicin (R), isoniazid (H), and pyrazinamide (Z), due to the lower likelihood of developing drug resistance compared to the other essential drugs [3]. In 2004, a worldwide study on drug resistance performed from 1994–1997 found that, in Brazil, the frequency of a primary isoniazid resistance was 5.9%, whereas of rifampicin was 1.1%, ethambutol 0.1%, streptomycin 6.5% and multidrug resistance was 0.9% [4]. In 2009, the WHO emphasized the previous recommendation, due to even higher rates of drug resistance reported in the late 90's in countries such as the Dominican Republic (8.6%) and India (13.3%) [5]. At that time, a second national survey on resistance to anti-tuberculosis drugs in Brazil showed an increase in isoniazid resistance alone (from 4.4 to 6.0%) or in its association with rifampicin (1.1 to 1.4%) [6]. For this reason, the Brazilian Ministry of Health, released a technical note changing the treatment regimen from RHZ to RHZE, making ethambutol as the fourth drug in intensive phase treatment, and decreasing the daily doses of isoniazid (from 400mg to 300mg, for an adult weighting more than 50 kg) and pyrazinamide (from 2000mg to 1600mg). Furthermore, the new guidelines changed the four-drugs presentation, combining them into one single tablet called a 'fixed dose combination' (RHEZ-FDC) [6,7].

Before the introduction of the new treatment regimen in 2009, a study that included 297 TB patients [8] reported an adverse event rate of 49.1% and 3.7% of patients required a change in therapy due to side effects. A different study in 519 patients from Germany reported an incidence rate of adverse events of 23% [9]. Moreover, the addition of a drug in an already complex therapeutic scheme could potentially increase the frequency of adverse drug reactions (ADRs) requiring interruption of treatment [10], contributing to increases in treatment dropouts and death, particularly in case of drug-induced hepatitis. [11]. Thus, safety of this regimen is a major concern for physicians and patients. After nearly six years after the implementation of the new therapeutic regimen in Brazil, it is necessary to evaluate the such change. The aim of this study was to determine the impact of moving from a three-drug to a four-drug regimen on cure and the frequency of ADR in TB patients from Brazil.

## Methods

### Ethics statement

All clinical investigations were conducted according to the principles set forth in the Declaration of Helsinki. The Ethics Committee of the Maternidade Climério de Oliveira, Federal

University of Bahia (UFBA), approved the study (protocol number: 1.092.460/2015-05). All information given to the research team were de-identified. The Ethics Committee exempted the patient's informed consent because the data were all from secondary sources.

## Design and procedures

The study was conducted with data from patients diagnosed with TB who received anti-TB treatment, between January 2007 and December 2014, at the Brazilian Institute for Tuberculosis Investigation (IBIT). This organization works as a referral center for TB treatment to 13–15% of patients with TB in Salvador/Bahia, city with approximately 3 million inhabitants located in Northeastern Brazil. This center maintains a high-quality standard of treatment and care of TB patients meeting the WHO recommendations on cure and abandonment rates [12]. Doses were administered according to WHO recommendations and Brazilian guidelines based on patient weight [13]

Data on clinical presentation, treatment outcomes, reported ADRs, demographic and laboratory characteristics (biochemical and microbiological) were obtained from medical records (raw data are shown in S1 File). In order to ensure complete data, patient registration and laboratory result logs from IBIT were reviewed. To ascertain accurate information on patients were cross-checked those with the national epidemiological surveillance system in Brazil databases: Notifiable Diseases Information System (Sistema de Informação de Agravos de Notificação–SINAN) and the Mortality Information System (Sistema de Informação de Mortalidade–SIM) [14,15].

## Sample size

Sample size was based on the expected 20% of adverse drug reactions rate for RHZ regimen and 30% for those with RHEZ regimen [11,16], assuming 80% power and a 95% confidence level. The estimated number was is 316 participants in each group, with a proportion of 1:1 (before and after drug regimen change) and estimating a safety margin of 15%, the estimated final number was 365 in each group. Randomization was performed using SPSS version 18. The power of the study after collecting data was 83.0%. The calculation of sample size and power were done using open source software, OpenEpi, version 3.01 [17].

## Eligibility criteria

The study included patients over 18 years of age, diagnosed with pulmonary or extra pulmonary TB between 2007–2014. The study period was defined based on the following reasons: (i) data from TB patients started being electronically captured at IBIT in 2007; (ii) the new regimen composed by 4-antitubercular drugs was implemented in Brazil by the Ministry of Health on 2014. Patients between January 2007 and February 2010 were included if they started a three-drug regimen [an example is $2R_{600}H_{400}Z_{2000}4R_{600}H_{400}/R_{600}H_{400}$ for patients over 45 kg (HR in FDC)] and between March 2010 and December 2014 if they started a four-drug regimen (an example is $2E_{1100}Z_{1600}4\ R_{600}H_{300}/R_{600}H_{300}$ for patients over 50 kg (RHEZ in FDC). Patients were ineligible if they were re-treated for TB or had a change in diagnosis during treatment, including a diagnosis of any drug resistance.

## Description of adverse drug reactions–ADR

An ADR was defined as "an unintended and harmful reaction to a medication, and which occurs at doses normally used in humans" [18]. An adverse event was defined as "any unfortunate medical event that may occur during treatment with a drug, but does not necessarily have

a causal relationship with this treatment" [18]. These two terms were used interchangeably in many cases. However, the final definition of ADR involved a degree of causality with the given treatment. In this paper, the ADRs were described according to the affected organ or system, following the Adverse Reaction Terminology (WHO-ART) [19]. The causality of ADR was assessed using the Naranjo Scale, as improbable = 0, possible = 1–4, probable = 5–8, definitely related ≥ 9 [20]. ADRs were classified as expected if the ADR was listed in the package insert for rifampicin, isoniazid, pyrazinamide and RHEZ-FDC [21–23].

## Statistical analysis

Demographic and clinical categorical variables were compared using a two-sided Pearson's chi-square test ($X^2$) (with Yates' correction) or two-tailed Fisher's exact test among 2X3 or 2x2 tables respectively. The Mantel-Haenszel Test (MH) made for comparison of patients with ADRs between treatment groups. Mean and standard deviation (SD) values were used as measures of central tendency and dispersion. Continuous variables were compared using the student's *t*-test, whereas the ANOVA test with Bonferroni post-hoc test was used to compare more than two groups. A multivariable regression model using variables with univariate p-value ≤ 0.2 was performed to assess the odds ratios (OR) and 95% confidence intervals (CIs) of the associations between the occurrence of at least one ADR and the exposure to RHEZ-FDC treatment adjusting for possible confounding factors. The p-values < 0.05 was considered statistically significant. Analyses were all pre-specified and were performed using SPSS version 21.

## Results

Between 2007 and 2014, data from patients with diagnosis of pulmonary or extra-pulmonary TB (with microbiological, histopathological or clinical and imaging-based diagnosis) [24,25] were extracted from medical records. After applying the eligibility criteria, there were 769 patients in the RHZ group and 1116 in the RHEZ-FDC group. To avoid sampling bias, we did randomization in order to assign 365 patients to each group (Fig 1).

Sex, pulmonary TB presentation, diabetes diagnosis, presentation of other comorbidities, concomitant medication use, current or past smoking and alcohol consumption were similar in both groups but mean age for the RHEZ-FDC group was higher than to the RHZ group (40.0 ± 13 years vs. 36.4 ± 13 years; p<0.01). There were no differences in the presentation of patients with MDR-TB between the groups of distinct anti-TB treatment (Table 1).

In this study, we did not classify ADRs by clinical severity, but the interruption of treatment suggested that the ADR was severe. Occurrence of treatment interruption and change in drug regimen schemes caused by ADR were similar between the two groups of patients stratified according to the distinct regimens. Frequency of patients who developed more than 1 episode of ADR was greater in the group of patients treated with RHEZ (53.5%) than in those who were treated with RHZ (32.7%). The causality was reported as possible in 97.1% of ADRs in the RHEZ group vs. 87.9% in the RHZ group (p<0.01) (Table 2).

ADR was more frequent in patients older than 30 years, especially in those using RHEZ-FDC (64 [74.4%] vs. 36 [65.5%]; p = 0.01) (Table 3). In the RHZ group, we found that 4 (1.1%) patients were simultaneously failure and MDR cases whereas in the RHEZ group, two patients (0.6%) had the co-condition, with no statistical difference between the groups (p = 0.68). Moreover, patients under RHEZ-FDC regimen developed more frequently ADRs than those using RHZ regimen (86 [23.6%] vs. 55 [15.1%]; p = 0.01) (Fig 2A). Cure, transfers, withdrawals and deaths were also similar between the groups (Fig 2B). There was one death in the RHEZ-FDC group related the regimen due to liver failure; this patient had also diabetes,

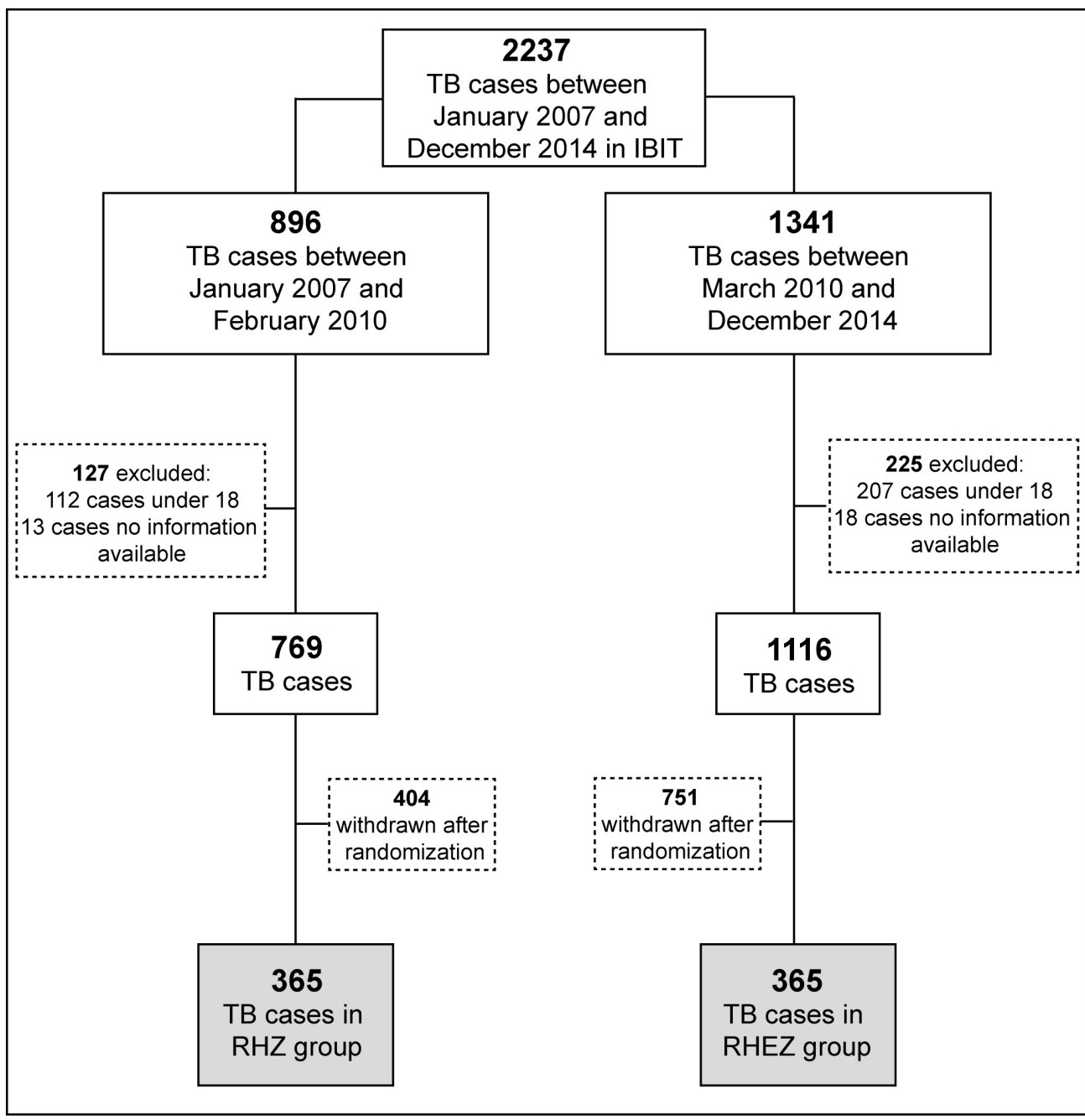

**Fig 1. Study flow diagram.** IBIT: Brazilian Institute for Tuberculosis Investigation (Reference Health Center); RHZ: Rifampicin (600mg), Isoniazid (400mg) and Pyrazinamide (2000mg). Dose to patients >45kg; RHEZ: Rifampicin (150mg), Isoniazid (75mg), Ethambutol (275mg) and Pyrazinamide (400mg) (fixed dose combination-FDC).

but no history of alcohol or tobacco consumption and the death occurred after 18 doses TB treatment. Another patient was hospitalized within 3 days due to the presentation of pruritus, vomiting diarrhea and melena; his clinical presentation was controlled during hospitalization and TB treatment resulted in microbiological and clinical cure.

The number of ADR episodes was greater in the four-drug regimen compared to three-drug group (170 [43.3%] vs. 107 [25.8%], respectively; Fig 2C). The most affected organ system

**Table 1. Baseline demographic and clinical presentation of tuberculosis patients.**

| Characteristics | Tuberculosis treatment | | p-value |
|---|---|---|---|
| | RHZ[1] (n = 365) | RHEZ-FDC[2] (n = 365) | |
| Male sex | 208 (57.0) | 200 (54.8) | 0.68 |
| Age in years | 40.0 ± 13 | 36 ± 13 | <0.01 |
| Pulmonary tuberculosis presentation[3] | 336 (92.1) | 324 (88.8) | 0.24 |
| Diabetes disease at diagnosis | 29 (4.9) | 33 (9.1) | 0.68 |
| Other comorbidities at diagnosis[4] | 28 (7.7) | 36 (9.9) | 0.30 |
| Concomitant medication[5] | 5 (1.4) | 9 (2.5) | 0.28 |
| Past or current cigarette smoking[6] | 77 (21.1) | 99 (27.1) | 0.06 |
| Past or current alcohol consumption[7] | 50 (13.7) | 67 (18.4) | 0.09 |
| MDR[8] | 16 (4.4) | 18 (4.9) | 0.86 |
| Patients with ADRs[9] | 55 (15.1) | 86 (23.6) | 0.01 |

Data represent no. (%). except age, represented in mean and standard deviation (SD)

[1]RHZ: Rifampicin (600mg), Isoniazid (400mg) and Pyrazinamide (2000mg). Dose to patients >45kg

[2]RHEZ: Rifampicin (150mg), Isoniazid (75mg), Ethambutol (275mg) and Pyrazinamide (400mg) (fixed dose combination-FDC). Dose to patients >50kg: 4 FDC.

[3]Pulmonary plus extra-pulmonary versus extra-pulmonary.

[4]Other comorbidities at diagnosis: stomach cancer, heart disease, mental disturbance, hypertension, bipolar disorder, psychotic outbreak, schizophrenia, rheumatoid arthritis, lymphedema, asthma, pyogenesis, pleural fibrosis, illicit drug use.

[5]Concomitant medication: Medicine taken concurrently with treatment anti-tuberculosis.

[6]Past or current cigarette smoker (daily use) versus no or sporadic use (non-smokers).

[7]Past or current alcohol consumption: regular (more than 3 days per week) versus no or sporadic–less than 3 days/week (non-drinkers).

[8]MDR: Multidrug resistance

[9]ADR: Adverse drug reactions

in both groups was gastrointestinal (RHEZ-FDC [27.6%] vs. RHZ [35.5%]; p = 0.17), followed by skin and appendages (RHEZ-FDC [24.1%] vs. RHZ [15.0%]; p = 0.07) (Fig 2D).

**Table 2. Distribution and characteristics of adverse drug reactions (ADRs) of tuberculosis patients.**

| ADR distribution and characteristics | Tuberculosis treatment | | p-value |
|---|---|---|---|
| | RHZ n = 55 | RHEZ-FDC n = 86 | |
| Patients with > 1 episode–n (%) | 18 (32.7) | 46 (53.5) | <0.01 |
| Treatment intrerruption[1] - n (%) | 6 (10.9) | 8 (9.3) | 0.14 |
| Treatment modifications[2] - n (%) | 2 (3.6) | 4 (4.6) | 0.79 |
| ADR characteristics[3] | | | |
| Expected ADRs—n (%) | 99 (92.5) | 165 (97.1) | 0.08 |
| Causality ADRs–n (%) | | | |
| Possible | 94 (87.9) | 165 (97.1) | <0.01 |
| Probable | 13 (12.1) | 4 (2.4) | <0.01 |

[1]The interruptions were due to toxicity, being two to four days, the minimum and maximum interruption time respectively.

[2]In the RHZ group, two modifications were for decreased dose isoniazid: 75mg/kg/day. In the group RHEZ, two modifications were for dose of isoniazid: 75mg/kg /day and two decreases dose isoniazid in 75 mg/kg/day and rifampicin at 150 mg/kg/day. All modifications were made after the reintroduction of the scheme with drugs as follows: rifampicin + ethambutol, followed by isoniazid and finally pyrazinamide.

[3]ADRs total in each group: n = 107 in RHZ group and n = 170 in RHEZ group

**Table 3. Treatment outcome and sputum smear result of tuberculosis patients.**

| Characteristics | Tuberculosis treatment | | | | | | |
|---|---|---|---|---|---|---|---|
| | RHZ | | | RHEZ-FDC | | | |
| | ADR n = 55 | No ADR n = 310 | p-value | ADR n = 86 | No ADR n = 279 | p-value | p-value* |
| **Age in years**** | 35 ± 11 | 37 ± 13 | 0.32 | 41 ± 12 | 39 ± 13 | 0.02 | 0.08 |
| **Range age ≥30 years** | 36 (65.5) | 191 (61.6) | 0.65 | 64 (74.4) | 197 (70.6) | 0.005 | 0.01 |
| **Female** | 32 (58.2) | 125 (40.3) | | 47 (54.7) | 118 (42.3) | | |
| **Pulmonary TB presentation** | 50 (90.9) | 286 (92.3) | 0.02 | 76 (88.4) | 248 (88.9) | 0.048 | 0.002 |
| **Treatment outcome** | | | 0.42 | | | 0.55 | 0.41 |
| Cure[1] | 47 (85.5) | 266 (85.8) | | 75 (87.2) | 251 (90) | | |
| Failure[2] | 0 (0) | 3 (1) | | 0 (0) | 0 (0) | | |
| Not evaluated[3] | 1 (1.7) | 0 (0) | | 3(3.5) | 0 (0) | | |
| Death[4] | 3 (5.5) | 6 (1.9) | | 2 (2.3) | 5 (1.8) | | |
| Lost to follow-up[5] | 4 (7.3) | 20 (6.5) | | 6 (7) | 10 (3.6) | | |
| **MDR[6]** | 1 (1.8) | 15 (4.8) | 0.48 | 3 (3.5) | 13 (4.7) | 0.78 | 0.49 |
| **Diabetes** | 4 (7.3) | 25 (8.1) | 1.00 | 14 (16.3) | 19 (6.8) | 0.16 | 0.06 |
| **Hypertension** | 1 (1.8) | 1 (0.3) | 0.28 | 2 (2.3) | 3 (1.1) | 0.34 | 0.14 |
| **Past or current alcohol consumption** | 18 (33.8) | 32 (2.3) | <0.01 | 15 (17.4) | 52 (18.6) | 0.78 | 0.08 |
| **Past or current cigarette smoking** | 19 (34.5) | 58 (18.7) | 0.02 | 25 (29.1) | 74 (26.5) | 0.78 | 0.09 |
| **Illicit drug use** | 1 (1.8) | 5 (1.6) | 1.00 | 3 (3.5) | 4 (1.4) | 0.38 | 0.29 |
| **Concomitant medication** | 3 (5.5) | 2 (0.6) | 0.03 | 3 (3.5) | 6 (2.2) | 0.46 | 0.04 |

Data represent no. (%), except age, represented in mean and standard deviation (SD)

* Mantel-Haenszel Test (MH) for comparison of patients with ADRs between treatment groups

** ANOVA for com for comparison of patients with ADRs between treatment groups

[1]Cured: A pulmonary TB patient with bacteriologically confirmed TB at the beginning of treatment who was smear- or culture-negative in the last month of treatment and on at least one previous occasion.

[2]Failure: Persistence of positive sputum at the end of treatment also classified as failure cases that at the beginning of treatment, are strongly positive (++ or +++) and maintain this situation until the fourth month, or those with initial positivity followed by negativity, and new positivity for two consecutive months, from the fourth month treatment.

[3]Not evaluated: A TB patient for whom no treatment outcome is assigned. This includes cases "transferred out" to another treatment unit.

[4]Death: A TB patient who dies for any reason before starting or during the course of treatment

[5]Lost to follow-up: A TB patient who did not start treatment or whose treatment was interrupted for two consecutive months or more

[6]MDR: Multidrug-resistant tuberculosis

The total number of cases MDR per year in Brazil increased between 2007 to 2014. In Salvador, a similar trend happened until 2014, where there was a decrease. At IBIT, the year 2014 presented the highest number of MDR cases (8 cases) between 2007 and 2014 (Fig 3A). Multivariable regression analysis revealed that TB patients with RHEZ-FDC treatment had 1.61 (95%CI: 1.10–2.35) times greater odds to have at least one adverse event (Fig 3A), after adjustment for age (≥ 30 years), alcohol consumption and diabetes (Fig 3B).

## Discussion

Although it is critical to assess the pros and cons of the implementation of the antitubercular treatment with RHEZ-FDC, there has no previous study directly comparing the current RHEZ-FDC therapy regimen with the previously used RHZ. The present study adds to the current knowledge in the field as it shows that the ADRs incidence increased after introduction of

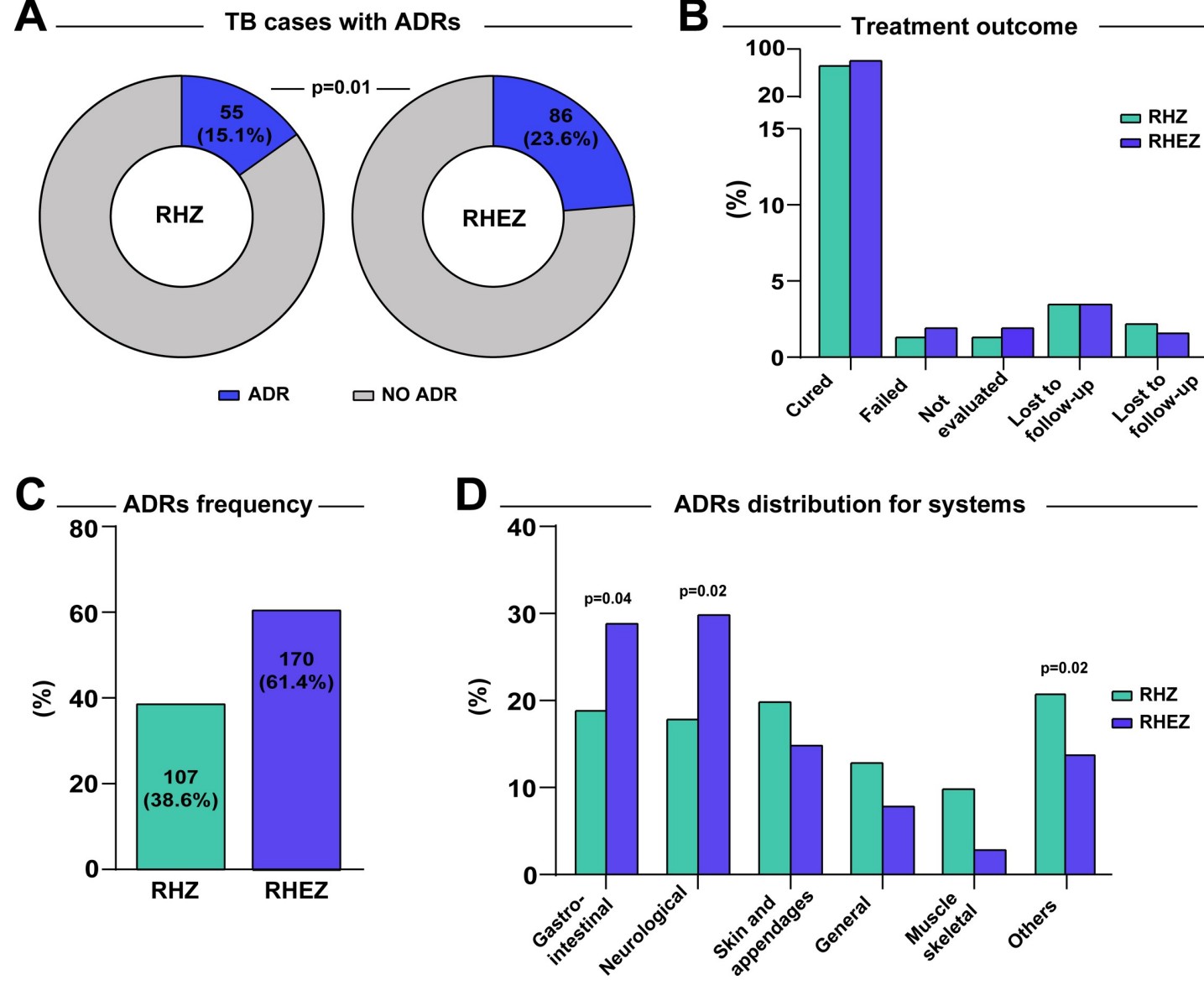

**Fig 2. Distribution of ADRs in TB patients.** (A) Frequency of ADRs in patients with RHZ and RHEZ anti- TB regime. (B) Outcomes treatment were compared between patients with RHZ and RHEZ anti-TB treatment. The significant p values are shown. (C) ADRs frequencies of anti-TB treatment groups with RHZ and RHEZ are shown. (D) ADRs distribution for systems were compared between patients with RHZ and RHEZ anti-TB treatment. The significant p-values are shown. Others: Hepatic, ocular, circulatory and respiratory system.

the fourth drug regimen in patients being treated for TB and this increase was more commonly observed after the third decade of age.

The proportion of participants presenting with at least one ADR event in this study was lower (24.2%) than what has been published by other studies performed in India [26], that reported an incidence of 74.1% and in Pakistan [27], in which the incidence reached up to 40.0%. In Brazil, a global incidence of patients with ADR is thought to be around 47% [28]. In our study, the frequency of ADRs within those under 30 years of age was similar in both groups, but the frequency increased significantly in the RHEZ-FDC group among those over 30 years of age, compared to that in those who received RHZ-based therapy. While there is

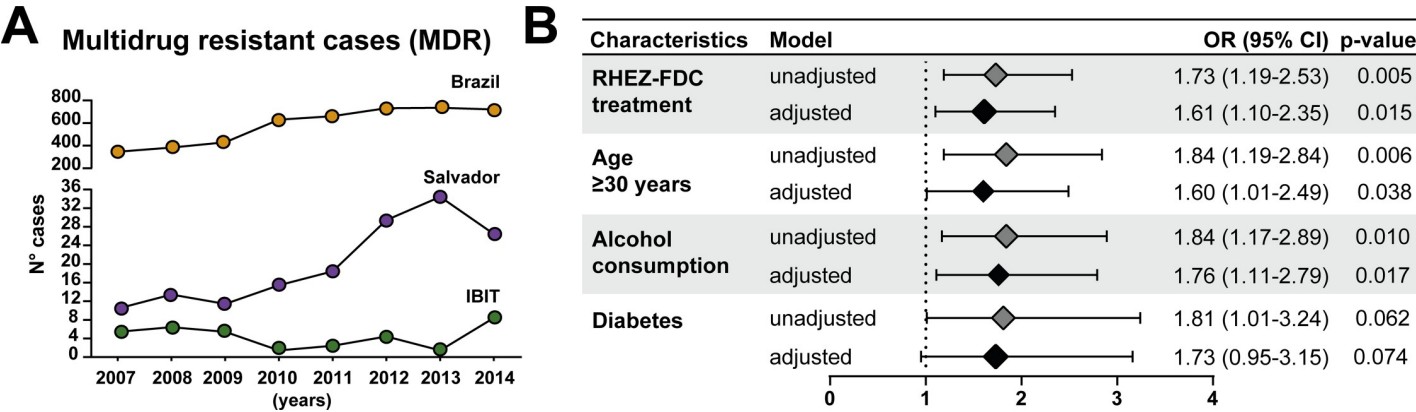

**Fig 3. MDR-TB cases and crude and odds ratio for adverse drug reaction in tuberculosis patients.** (A) Total number of MDR-TB cases shown per year in Brazil, Salvador and IBIT (B) A multivariable regression model adjusted for RHEZ-FDC treatment, age ≥ 30 years, alcohol consumption and diabetes. The odds associated with the covariates used in the model adjustment are displayed in Table 3. OR = Odds ratio; CI = confidence intervals.

evidence to support the idea that ADR events may be more frequent in TB patients proportional to age [29], other studies found no association between increasing age and risk of adverse reactions [30]. The latter study justified its results speculating that physicians prescribed more often hepatoprotective drugs, such as N-acetyl-cysteine, during the treatment of elderly patients. Patients included in the group from 2010 to 2014 and who were treated with the four-drug regimen were significantly older compared with those from the previous years and who received three drugs. This difference could be attributed to the accelerated aging of the Brazilian population, also described in people with TB in those years [31]. Furthermore, epidemiological factors such as TB and HIV control programs could have impacted the TB prevalence in age groups below 30 years old [32].

Our findings indicate that the most common body systems affected by the four-drug regimen were the gastrointestinal tract and neurological. Hence, the most frequent ADR was vomiting, which can be considered from mild to moderate, and managed symptomatically without interruption of anti TB treatment [33]. In the RHZ group, the most affected system were gastrointestinal and neurological and for RHEZ -FDC group was the skin and appendages, which is expected due to the use of pyrazinamide and ethambutol, that are responsible for skin reactions [34]. Our results differ from those reported in a study from China [30], which found that the most frequent ADRs occurred in the liver, representing 9.3% of all the ADRs, while, in our study, the incidence of liver-related ADR was less than 1%. This result could be attributed to underreporting of ADRs in our study, and that not all patients had laboratory results of the liver function tests during treatment.

Previous studies have shown that alcohol and tobacco consumption may predispose and accelerate hepatotoxic effects, caused especially by isoniazid [29,30]. In the present study, we found that the regimen with RHEZ was strongly associated with occurrence of at least one ADR event independent of other confounding factors. Nevertheless, risk variables such as malnutrition or history of liver problems could not be obtained since the data were not recorded in all cases.

Of note, herein no ADR event was classified as definitive. Most of the ADRs have been previously described in the medication's package insert and only 1.9% required the discontinuation of treatment as a result of the ADR, which was lower than what it has been reported by a similar study [35]. In this context, the presentation of ADR becomes a challenge for TB control that has not yet been completely deluded. It is currently known that there are crucial factors

associated with ADRs, such as genetic variants, which influence the effectiveness of treatment and the presentation of ADR [36,37]. Enzymes that metabolize anti-TB drugs can undergo genetic or epigenetic alterations, which may result in reduced or complete loss of activity, thus causing toxic damage to cells [36,37].

RHEZ-FDC have been implemented for those under basic drug-sensitive anti-tuberculosis treatment regimen, to both reduce the number of MDR-TB cases and decrease poor outcomes of treatment [38]. However, there was no major change in the number cases MDR-TB in Brazil at the time of implementation of the new four-drug regimen. Although FDC was recommended by the WHO and IUTLD to reduce emergence of drug resistance, other studies have suggested that the use of FDC is related to the risk of treatment failure or TB recurrence, depending on the bioavailability of drugs [2]. Thus, the fact that FDC implementation had no effect on the change in MDR-TB cases in Brazil could have been due to the implementation of fixed-dose use in anti-TB treatment without the evaluation of the bioavailability of drugs, especially rifampicin, as emphasized when the recommendation was published [2]. Furthermore, the addition of ethambutol to the RHZ-FDC scheme could have reduced the risk of additional resistance to isoniazid monoresistance, rifampin or simultaneous resistance to isoniazid and pyrazinamide, but not in cases of simultaneous resistance to isoniazid and rifampicin, which defines MDR cases [39]. Our results together with those of other studies [11,39] suggest the great need to perform the bioavailability assessment and also the sensitivity tests during implementation of FDC.

Our results show differences in the trend of reported cases such as MDR-TB between the Salvador city and IBIT through years 2007–2014, this could be explained by the holistic management, performed at IBIT with services offered in conjunction with specialized mental health centers and psychosocial care centers as well as continuous patient support [12].

This study shows that failure outcomes in both treatments were similar, but this study was not designed to assess outcomes of anti-TB treatment. It is most probably too early to observe the effect on efficacy however, the efficacy of RHEZ-FDC has not yet been fully established as observed in a recent systematic review [40].

There are several limitations in our study. When using secondary data, these may be inaccurate. As indicated earlier, the patients in this study were treated with several medications, therefore, it is difficult to analyze the causality between the presentation of ADR and the specific medications separately. Another important point is that not all participants attended monthly clinical evaluations or were not tested in the laboratory during treatment, which could carry out a sub-notification of ADR or not obtain more information to establish the intensity/severity of ADR.

In conclusion, our findings indicated that the risk of ADR is likely more frequent in TB patients receiving RHEZ-FDC compared to that in those undertaking RHZ. This difference was more striking among those over 30 years of age. Such high-risk patient subpopulation should be followed up more carefully when the 4-drug regimen is being used [41].

## Supporting information

**S1 File. De-identified raw data extracted from the medical records.**
(XLSX)

## Acknowledgments

We thank Fundação José Silveira for the support.

## Author Contributions

**Conceptualization:** María B. Arriaga, Ninfa M. C. Torres, Eduardo M. Netto.

**Formal analysis:** María B. Arriaga, Eduardo M. Netto.

**Investigation:** Eduardo M. Netto.

**Methodology:** Nelia C. N. Araujo, Simone C. C. Caldas, Eduardo M. Netto.

**Supervision:** Ninfa M. C. Torres, Bruno B. Andrade.

**Writing – original draft:** María B. Arriaga, Ninfa M. C. Torres, Bruno B. Andrade, Eduardo M. Netto.

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
