## [Decision Letter · Decision Letter 0]

6 Nov 2019

PONE-D-19-25724

Impact of the change in the antitubercular regimen from three to four drugs on cure and frequency of adverse reactions in tuberculosis patients from Brazil: a retrospective cohort study

PLOS ONE

Dear Dr. Andrade,

Thank you for submitting your manuscript to PLOS ONE. After careful consideration, we feel that it has merit but does not fully meet PLOS ONE’s publication criteria as it currently stands. Therefore, we invite you to submit a revised version of the manuscript that addresses the points raised during the review process.

Please see list of suggestions below, please provide a revised paper with a point by point rebuttal letter

We would appreciate receiving your revised manuscript by Dec 21 2019 11:59PM. To enhance the reproducibility of your results, we recommend that if applicable you deposit your laboratory protocols in protocols.io, where a protocol can be assigned its own identifier (DOI) such that it can be cited independently in the future. For instructions see: http://journals.plos.org/plosone/s/submission-guidelines#loc-laboratory-protocols

We look forward to receiving your revised manuscript.

Kind regards,

Christophe Sola, Pharm.D., Ph.D.

Academic Editor

PLOS ONE

Journal Requirements:

1. Please refer to any post-hoc corrections made during your statistical analysis. Please justify the reasons if these were not performed. Additionally, please thoroughly discuss any limitations of your study within the Discussion section.

2. Please include a copy of Table 4 which you refer to in your text on page 10

Additional Editor Comments (if provided):

PONE-D-19-25724 Review reviewer 1

In this work Arriaga et al report changes occurring in the overall outcome of patients and in the frequency of adverse reactions subsequent to the modification of the antitubercular regimen from three to four drugs in Brazil. This a retrospective cohort study which enrolled 365 patients treated with three anti-TB drug regimen (RHZ) from the Jan 2007 to Feb 2010 and 365 patients treated with four anti-TB drug regimen (RHEZ) from Mar 2010 to Dec 2014.

The main results are an increase in the adverse effects which were higher in the RHEZ (23.6%) vs RHZ (15.1%), especially in individuals > 30 years. There were no significant differences between the two groups regarding the outcome (cure or failure), though the authors stress the fact that the study was not designed to point to such differences.

Altogether this is an interesting report on the consequences of the adjustment of the standard recommended anti-TB regimen, capturing an extended period of time and a large retrospective cohort. However, several aspects need clarification prior to publication.

Comments:

1. Page 3 line 46: isoniazid instead of rifampicin

2. Adverse effects (ADR) were higher in the RHEZ (23.6%) vs RHZ (15.1%), especially in individuals > 30 years. However, it should be stressed that severe ADR leading to either interruption or treatment modification was similar between the two regimens.

3. Though authors indicate that failure rates were unchanged, failure seems to have occurred only in the RHZ group. Could the authors clarify this issue? Were the failures related to resistance emergence?

4. Are data regarding the evolution of isoniazid mono-resistance concomitant with the regimen change available?

5. Which were the neurological side effects observed? Was-it ethambutol-induced optic neuropathy?(less...)

PONE-D-19-25724 Review reviewer 2

General comments

This retrospective study assess the ADR of two drug regimens used for tuberculosis treatment. It is a topic of high societal interest, done on a significant number of patients, however with some points that could be improved. The English language is not optimal and the paper should be reviewed by a native English speaker. Moreover the discussion could be substantially improved.

Specific comments

“Since 1994, the WHO, together with the International Union Against Tuberculosis and Lung Disease (IUATLD) recommended the use of fixed dose combination (FDC) in treatment of TB because it simplifies drug dispensing and prevents the development of drug resistance [2,3].”

According to the reviewer, this might be an historical mistake: FDC started to be recomemmended by WHO only in 2001 and not in 1994 when TB was declared as World-wide emergency (1993). Please check and provide reference.

Bull World Health Organ. 2001;79(1):61-8. Epub 2003 Nov 5. The rationale for recommending fixed-dose combination tablets for treatment of tuberculosis. Blomberg B1, Spinaci S, Fourie B, Laing R

Lane 55 : please already mention here the most important ADR observed, related to treatment changes.

Lane 111 : please explain if there was a stratification between Adverse events non requiring hospitalization (minor, mild or moderate) from Adverse events requiring hospitalization (major, severe). Comment if not and in particular try to evaluate if some ADR required hospitalization or not, if not mention it.

Lane 145: mean age for the RHEZ-FDC group was higher than to the RHZ group (40.0 ± 13 years vs. 36.4 ± 13 years; p<0.01).

Please explain why, if randomly sampled, there is a significant difference of age between the two groups, in particular comment if this difference could be related to and decrease TB recent transmission rate relative to reactivation in group RHEZ versus to group RHZ.

Lane 196-198 : please comment on the reasons of the observed discrepancy between IBIT figures and Salvador figures

Lane 239. :”field” should replace “filed”

Lane 292 : “though there was no major change in the number cases MDR-TB in Brazil at the time of implementation of the new four-drug regimen”.

I think that this point deserves more discussion, indeed, if the cure rate is not significantly different in the new regimen (point not so well discussed and difficult to estimate on Fig 2B) , if there are more ADR and if MDR-TB rate does not descrease, then what is the true impact of this new regimen as mentioned two lines after ?

The paper also mentions a Table 4 that was not found and the reference to this Table should be suppressed

Reviewers' comments:

Reviewer's Responses to Questions

**Comments to the Author**

1. Is the manuscript technically sound, and do the data support the conclusions?

Reviewer #1: Yes

2. Has the statistical analysis been performed appropriately and rigorously? 

Reviewer #1: Yes

3. Have the authors made all data underlying the findings in their manuscript fully available?

Reviewer #1: Yes

4. Is the manuscript presented in an intelligible fashion and written in standard English?

Reviewer #1: Yes

5. Review Comments to the Author

Reviewer #1: In this work Arriaga et al report changes occurring in the overall outcome of patients and in the frequency of adverse reactions subsequent to the modification of the antitubercular regimen from three to four drugs in Brazil. This a retrospective cohort study which enrolled 365 patients treated with three anti-TB drug regimen (RHZ) from the Jan 2007 to Feb 2010 and 365 patients treated with four anti-TB drug regimen (RHEZ) from Mar 2010 to Dec 2014.

The main results are an increase in the adverse effects which were higher in the RHEZ (23.6%) vs RHZ (15.1%), especially in individuals > 30 years. There were no significant differences between the two groups regarding the outcome (cure or failure), though the authors stress the fact that the study was not designed to point to such differences.

Altogether this is an interesting report on the consequences of the adjustment of the standard recommended anti-TB regimen, capturing an extended period of time and a large retrospective cohort. However, several aspects need clarification prior to publication.

Comments:

1. Page 3 line 46: isoniazid instead of rifampicin

2. Adverse effects (ADR) were higher in the RHEZ (23.6%) vs RHZ (15.1%), especially in individuals > 30 years. However, it should be stressed that severe ADR leading to either interruption or treatment modification was similar between the two regimens.

3. Though authors indicate that failure rates were unchanged, failure seems to have occurred only in the RHZ group. Could the authors clarify this issue? Were the failures related to resistance emergence?

4. Are data regarding the evolution of isoniazid mono-resistance concomitant with the regimen change available?

5. Which were the neurological side effects observed? Was-it ethambutol-induced optic neuropathy?

6. PLOS authors have the option to publish the peer review history of their article (what does this mean?). If published, this will include your full peer review and any attached files.

Reviewer #1: No

---

## [Author Response · Author response to Decision Letter 0]

20 Nov 2019

PONE-D-19-25724 Review reviewer 1

Reviewer #1: In this work Arriaga et al report changes occurring in the overall outcome of patients and in the frequency of adverse reactions subsequent to the modification of the antitubercular regimen from three to four drugs in Brazil. This a retrospective cohort study which enrolled 365 patients treated with three anti-TB drug regimen (RHZ) from the Jan 2007 to Feb 2010 and 365 patients treated with four anti-TB drug regimen (RHEZ) from Mar 2010 to Dec 2014.

The main results are an increase in the adverse effects which were higher in the RHEZ (23.6%) vs RHZ (15.1%), especially in individuals > 30 years. There were no significant differences between the two groups regarding the outcome (cure or failure), though the authors stress the fact that the study was not designed to point to such differences.

Altogether this is an interesting report on the consequences of the adjustment of the standard recommended anti-TB regimen, capturing an extended period of time and a large retrospective cohort. However, several aspects need clarification prior to publication.

Reply: We thank the reviewer for his constructive criticism, which has led to significant improvement of the manuscript.

Comments:

General comments

1. Please refer to any post-hoc corrections made during your statistical analysis. Please justify the reasons if these were not performed. 

Reply: Thanks for your observation. The following tests were performed: One-Way ANOVA with Bonferroni post-hoc test was used to compare continuous variables between more than two groups. In addition, Pearson’s chi-square test (with Yates’ correction) or two-tailed Fisher’s exact test was used to compare categorical variables among 2X3 or 2x2 tables respectively. We have now updated the statistical analysis section in the methodology and in the table 3.

2. Additionally, please thoroughly discuss any limitations of your study within the Discussion section.

Reply: The text of the limitations was added to the discussion section.

3. Please include a copy of Table 4 which you refer to in your text on page 10

Reply: The text was supposed to refer to Figure 3B instead of Table 4; it was a typo caused by reformulation of a previous version of the manuscript. We apologize for the typo which has now been corrected.

Additional Editor Comments (if provided):

PONE-D-19-25724 Review reviewer 1

In this work Arriaga et al report changes occurring in the overall outcome of patients and in the frequency of adverse reactions subsequent to the modification of the antitubercular regimen from three to four drugs in Brazil. This a retrospective cohort study which enrolled 365 patients treated with three anti-TB drug regimen (RHZ) from the Jan 2007 to Feb 2010 and 365 patients treated with four anti-TB drug regimen (RHEZ) from Mar 2010 to Dec 2014.

The main results are an increase in the adverse effects which were higher in the RHEZ (23.6%) vs RHZ (15.1%), especially in individuals > 30 years. There were no significant differences between the two groups regarding the outcome (cure or failure), though the authors stress the fact that the study was not designed to point to such differences.

Altogether this is an interesting report on the consequences of the adjustment of the standard recommended anti-TB regimen, capturing an extended period of time and a large retrospective cohort. However, several aspects need clarification prior to publication.

Comments:

1. Page 3 line 46: isoniazid instead of rifampicin

Reply: We have made the indicated change accordingly.

2. Adverse effects (ADR) were higher in the RHEZ (23.6%) vs RHZ (15.1%), especially in individuals > 30 years. However, it should be stressed that severe ADR leading to either interruption or treatment modification was similar between the two regimens.

Reply: The text was updated in results section accordingly.

3. Though authors indicate that failure rates were unchanged, failure seems to have occurred only in the RHZ group. Could the authors clarify this issue? Were the failures related to resistance emergence?

Reply: Yes. A hypothesis is that there is a relationship between occurrence of treatment failure and the emerging number of drug resistant cases. In fact, in the RHZ group, we found that 4 patients were simultaneously failure and MDR cases whereas in the RHEZ two patients had the co-condition. We have added this information to the Results section (xxx). 

Are data regarding the evolution of isoniazid mono-resistance concomitant with the regimen change available?

Reply: No was concomitant, since the change was made since March 2010 and the primary resistance to H increased from 4.4% (period 1995-1997) to 6.0% (2007-2008), in Brazil

4. Which were the neurological side effects observed? Was-it ethambutol-induced optic neuropathy?(less...)

Reply: In both groups they presented: migraine, headache and stinging. Only one case and ocular neuropathy occurred in the RHEZ-FD group, but no causality could be established because the patient did not have other clinical evaluations in the center and was transferred to a referral hospital.

We have added the requested information following the reviewer’s recommendation

PONE-D-19-25724 Review reviewer 2

General comments

This retrospective study assess the ADR of two drug regimens used for tuberculosis treatment. It is a topic of high societal interest, done on a significant number of patients, however with some points that could be improved. The English language is not optimal and the paper should be reviewed by a native English speaker. Moreover the discussion could be substantially improved.

Reply: The entire text was reviewed by a native English speaker and the discussion was updated.

Specific comments

1. “Since 1994, the WHO, together with the International Union Against Tuberculosis and Lung Disease (IUATLD) recommended the use of fixed dose combination (FDC) in treatment of TB because it simplifies drug dispensing and prevents the development of drug resistance [2,3].”

According to the reviewer, this might be an historical mistake: FDC started to be recommended by WHO only in 2001 and not in 1994 when TB was declared as World-wide emergency (1993). Please check and provide reference: Bull World Health Organ. 2001;79(1):61-8. Epub 2003 Nov 5. The rationale for recommending fixed-dose combination tablets for treatment of tuberculosis. Blomberg B1, Spinaci S, Fourie B, Laing R

Reply: The document entitled “The promise and the reality of fixed-dose combinations with rifampicin. A joint statement of the International Union Against Tuberculosis and Lung Diseases and the Tuberculosis Programme of the World Health Organization. International Journal of Tuberculosis and Lung Disease, 1994, 75: 180–181” describes that the first recommendation was made by IUATLD in 1988 and that the WHO had emphasized such recommendation in 1993. 

2. Lane 55 : please already mention here the most important ADR observed, related to treatment changes.

Reply: We have now added to the indicated text the most important ADR observed, which was migraine.

3. Lane 111: please explain if there was a stratification between Adverse events non requiring hospitalization (minor, mild or moderate) from Adverse events requiring hospitalization (major, severe). Comment if not and in particular try to evaluate if some ADR required hospitalization or not, if not mention it.

Reply: Only one patient reported having been hospitalized. We have added this information to the text in the Results section.

4. Lane 145: mean age for the RHEZ-FDC group was higher than to the RHZ group (40.0 ± 13 years vs. 36.4 ± 13 years; p<0.01). Please explain why, if randomly sampled, there is a significant difference of age between the two groups, in particular comment if this difference could be related to and decrease TB recent transmission rate relative to reactivation in group RHEZ versus to group RHZ.

Reply: In the period 2010 and 2014 (DATASUS-SINAN), there was an age-specific change in TB incidence. There was an increase in the median age of cases of all forms of TB and in the proportion of cases among older individuals and adults aged 30 years or older. These trends have possibly been determined by demographic factors (population growth and aging) as well as epidemiological factors (TB control programs, HIV infection). (Gaspar RS et al. J. Bras. Pneumol. 2016, vol.42, n.6, pp.416-422. SINAN - Sistema de Informação de Agravos de Notificação [Internet]. Available: http://sinan.saude.gov.br/sinan/login/login.jsf)

In addition, the drop in fertility rates that began in the mid-1960s has kept the absolute number of young people constant. This situation may have contributed to the stability in the number of TB cases in 0-39yo age groups.

Another interesting point is the improvement of tuberculosis control programs in order to reduce the annual risk of infection, which has led to a decrease in the incidence of TB observed in groups less than 30 years old. We have added this rationale in the Discussion section (line 267).

5. Lane 196-198: please comment on the reasons of the observed discrepancy between IBIT figures and Salvador figures

Reply: There are several risk factors for the development of MDR TB such as psychiatric diseases, alcoholism, drug addiction and homelessness which can predict non-adherence to treatment. Poor compliance with treatment is also an important factor in the development of acquired drug resistance. All these factors have been addressed at IBIT, with comprehensive services offered in conjunction with specialized mental health centers and psychosocial care centers as well as continuous patient support. This holistic health management performed at IBIT could be related to a discrepant results observed in Figure 3A. (Slatery, Erin, "Neglected or Non-compliant? Assessing the difficulties of tuberculosis patients in Salvador-BA, Brazil" (2014). Independent Study Project (ISP) Collection.1944. https://digitalcollections.sit.edu/isp_collection/1944). We have added this hypothesis to the Discussion section (line 318). 

6. Lane 239 :”field” should replace “filed”

Reply: We have made the indicated change

7. Lane 292 : “though there was no major change in the number cases MDR-TB in Brazil at the time of implementation of the new four-drug regimen”. 

I think that this point deserves more discussion, indeed, if the cure rate is not significantly different in the new regimen (point not so well discussed and difficult to estimate on Fig 2B), if there are more ADR and if MDR-TB rate does not descrease, then what is the true impact of this new regimen as mentioned two lines after ?

Reply: The discussion section was updated with the reviewer's observation (line 304)

8. The paper also mentions a Table 4 that was not found and the reference to this Table should be suppressed

Reply: We apologize for the typo, which was meant to cite Figure 3B instead of Table 4, that does not exist.

---

## [Editor Report · Decision Letter 1]

13 Dec 2019

Impact of the change in the antitubercular regimen from three to four drugs on cure and frequency of adverse reactions in tuberculosis patients from Brazil: a retrospective cohort study

PONE-D-19-25724R1

Dear Dr. Andrade,

We are pleased to inform you that your manuscript has been judged scientifically suitable for publication and will be formally accepted for publication once it complies with all outstanding technical requirements.

With kind regards,

Christophe Sola, Pharm.D., Ph.D.

Academic Editor

PLOS ONE
---

## [Editor Report · Acceptance letter]

17 Dec 2019

PONE-D-19-25724R1 

Impact of the change in the antitubercular regimen from three to four drugs on cure and frequency of adverse reactions in tuberculosis patients from Brazil: a retrospective cohort study 

Dear Dr. Andrade:

I am pleased to inform you that your manuscript has been deemed suitable for publication in PLOS ONE. Congratulations! Your manuscript is now with our production department. 

With kind regards,

on behalf of

Pr. Christophe Sola 

Academic Editor

PLOS ONE